# Genetic Characterization of Non-*Lymphogranuloma venereum Chlamydia trachomatis* Indicates Distinct Infection Transmission Networks in Spain

**DOI:** 10.3390/ijms24086941

**Published:** 2023-04-08

**Authors:** Luis Piñeiro, Laura Villa, Paula Salmerón, Maria Dolores Maciá, Luis Otero, Martí Vall-Mayans, Ana Milagro, Samuel Bernal, Ayla Manzanal, Iñigo Ansa, Gustavo Cilla

**Affiliations:** 1Microbiology Department, Donostia University Hospital-Biodonostia Health Research Institute, 20014 San Sebastian, Spain; 2Sexually Transmitted Infections Study Group of the Infectious Diseases and Clinical Microbiology Spanish Society (GEITS-SEIMC), 28003 Madrid, Spain; 3Microbiology Department, Central University Hospital of Asturias, 33011 Oviedo, Spain; 4Microbiology Department, Vall d’Hebrón University Hospital, 08035 Barcelona, Spain; 5Microbiology Department, Son Espases University Hospital, 07120 Palma de Mallorca, Spain; 6Microbiology Department, Cabueñes University Hospital, 33394 Gijón, Spain; 7Vall’Hebron-Drassanes STI Unit, Infectious Diseases Department, Vall d’Hebrón University Hospital, 08035 Barcelona, Spain; 8Microbiology Department, Miguel Servet University Hospital, 50009 Zaragoza, Spain; 9Infectious Diseases and Microbiology Unit, Valme University Hospital, 41014 Seville, Spain

**Keywords:** *Chlamydia trachomatis*, *ompA* genotyping, multilocus sequence typing, sexual behaviour

## Abstract

*Chlamydia trachomatis* infection is an important public health problem. Our objective was to assess the dynamics of the transmission of this infection, analysing the distribution of circulating *ompA* genotypes and multilocus sequence types of *C. trachomatis* in Spain as a function of clinical and epidemiological variables. During 2018 and 2019, we genetically characterized *C. trachomatis* in tertiary hospitals in six areas in Spain (Asturias, Barcelona, Gipuzkoa, Mallorca, Seville and Zaragoza), with a catchment population of 3.050 million people. Genotypes and sequence types were obtained using polymerase chain reaction techniques that amplify a fragment of the *ompA* gene, and five highly variable genes (*hctB*, CT058, CT144, CT172 and *pbpB*), respectively. Amplicons were sequenced and phylogenetic analysis was conducted. We obtained genotypes in 636/698 cases (91.1%). Overall and by area, genotype E was the most common (35%). Stratifying by sex, genotypes D and G were more common among men, and genotypes F and I among women (*p* < 0.05). Genotypes D, G and J were more common in men who have sex with men (MSM) than in men who have sex with women (MSW), in whom the most common genotypes were E and F. The diversity index was higher in sequence typing (0.981) than in genotyping (0.791), and the most common sequence types were ST52 and ST108 in MSM, and ST30, ST148, ST276 and ST327 in MSW. Differences in genotype distribution between geographical areas were attributable to differences in population characteristics. The transmission dynamics varied with sexual behaviour: the predominant genotypes and most frequent sequence types found in MSM were different to those detected in MSW and women.

## 1. Introduction

*Chlamydia trachomatis* is the leading bacterial cause of sexually transmitted infections (STIs) and the cause of trachoma. It is estimated that there are some 127 million new cases of *C. trachomatis* every year worldwide [1]. In Europe, the incidence of *C. trachomatis* STIs is high (157/100,000 in 2019, 2020 and 2021 being two years with a lower epidemiological statement due to the COVID pandemic), particularly in young people and men who have sex with men (MSM) (disease data from ECDC Surveillance Atlas: Chlamydia; https://europa.eu, accessed on 4 April 2023). Given this situation and the potential consequences of this genital infection, including prostatitis, epididymitis, pelvic inflammatory disease, infertility and pregnancy complications, as well as neonatal infections, the European Centre for Disease Prevention and Control has encouraged European countries to implement measures to improve its control [2]. Regional studies on molecular epidemiology of the infection enable us to improve our understanding of transmission patterns and identify high-risk sexual networks [3], which helps guide options for preventive strategies.

Based on the antigenicity of their major outer membrane protein (MOMP) encoded by the *ompA* gene, strains of *C. trachomatis* can be classified into genotypes, primarily associated with trachoma (genotypes A–C), oculogenital tract infections (genotypes D–K) or lymphogranuloma venereum (LGV, genotypes L1–L3). Among the approaches used to analyse the molecular epidemiology of *C. trachomatis, ompA* genotyping has been the most widely used. On the other hand, multilocus sequence typing (MLST), a high-resolution technique analysing several genes, is better for discriminating between strains than *ompA* genotyping, and is less influenced by genetic recombination, a major mechanism for genetic variation and evolution in this bacterial species [4,5]. In the context of *C. trachomatis* infections, the MLST has proven to be a good tool for epidemiological studies. Other techniques used for the molecular characterization of *C. trachomatis* are whole-genome sequencing and the analysis of tissue-tropism-associated genes located in the so-called plasticity zone of the genome [6,7,8,9].

In Spain, infection by *C. trachomatis* was included in the Notifiable Diseases Surveillance System in 2015, the resulting reports helping obtain a better picture of its epidemiology [10]. On the other hand, there are limited data on the molecular epidemiology of non-LGV *C. trachomatis* [11,12,13]. To help fill the knowledge gaps, this study had two objectives: (1) to analyse the *ompA* genotypes of non-LGV *C. trachomatis* to determine their distribution geographically across Spain, and as a function of certain epidemiological characteristics of the population (including age, sex and sexual behaviour), as well as genotypic changes over time; and (2) to carry out MLST in a selection of samples covering all the genotypes, considering the aforementioned variables, and analyse the potential relationship of sequence types (STs) with transmission dynamics and tissue tropism.

## 2. Results

### 2.1. Genotyping

Over the 2 years of the study, a total of 698 *C. trachomatis*-positive samples were selected, and *ompA* genotypes were determined for 636 samples (91.1%). There were some clinical and epidemiological differences between the populations analysed in the different areas, including higher percentages of women in Asturias and MSM in Seville and Barcelona (Table 1, Appendix A).

We detected genotypes D–K (DI = 0.791), widely spread across the geographical areas sampled. All the genotypes were observed in four out of the six areas studied, while seven and six of the genotypes were found in samples from the other two areas. Genotype E was the most common overall (35.4%), followed by D, F and G (together overall 46.4%), while J, I, H and K were the least common, accounting for just 17.7%. Lastly, we also found three samples containing genotype B (0.5%), which—though all from the same area (Asturias)—were apparently epidemiologically unrelated.

We did not find any differences in the distribution of genotypes by age group (<25 vs. ≥25 years old), country of origin (Spanish vs. foreign-born), or whether patients experienced symptoms. In contrast, we did find differences by sex and sexual behaviour (Table 2). Genotypes D and G were more common among men than women, while genotypes F and I were more common among women than men (*p* < 0.05). Among men, genotypes D, G and J were more common in MSM, accounting for 78.7% of cases (*p* < 0.01), while genotypes E and F were more common in MSW, accounting for 60.0% (*p* < 0.01). Notably, genotype E represented only 14% of the strains genotyped in MSM, but 40.1% of those identified in the rest of the population (*p* < 0.01).

### 2.2. Sequence Typing

Based on the aforementioned genotype distribution analysis, 158 samples were selected for MLST, with a lower representation of genotypes E and F, to facilitate the analysis of potential differences in STs in genotypes D, G and J as a function of sexual behaviour that might explain differing patterns of transmission, as well as to improve our understanding of STs in the least common genotypes (Table 3, Appendix A). We obtained STs from 142 samples (89.9%), corresponding to 67 different STs (DI = 0.981), of which 20 (29.9%) had not been described previously (18 with new combinations of the five alleles and two containing new *pmpB* alleles, ST 586 and ST 589) and have been allocated ST numbers 571–590 (https://pubmlst.org, accessed on 4 April 2023). All the *ompA* genotypes were genetically diverse, 36.8–62.5% of the intra-genotype strains characterised corresponding to different STs (*p* = NS), and new STs being detected in genotypes D–K. Nonetheless, some sequence types were particularly successful, accounting for more than 25% of the strains of the corresponding *ompA* genotype, as in the cases of ST35 (genotype D, n = 6/23), ST148 (genotype F, n = 5/14), ST52 (genotype G, n = 10/31), ST108 (genotype J, n = 5/17), STs 276 and 100 (genotype I, n = 7/19 and 6/19, respectively), ST97 (genotype H, n = 6/10) and ST30 (genotype K, n = 4/10), with no geographical clustering (i.e., these STs being detected in at least two areas). On the other hand, some STs (12, 56, 90, 108 and 275) were detected in two genotypes. Further, ST58 was detected in two samples containing genotype D that showed LGV coinfection.

Given the large number of different STs, we were not able to analyse the differences in their distribution as a function of all of the variables studied, but we did observe a significant association (*p* = 0.034) between ST distribution and sexual behaviour with STs 30, 35, 97, 100, 128, 148, 276 and 327 detected in women and MSW, and STs 33, 52, 108, 109 and 571 in MSM (Appendix A). Although the DI was higher in MSM than in MSW (0.977 vs. 0.947), the difference in clonality (number of STs/genotype) between the two groups did not reach statistical significance.

Finally, the distribution of STs also differed between rectal samples from women (four samples each containing one ST, namely, ST 100, 128, 147, or 276) and MSM (23 samples containing STs 33, 52, 56, 58, 97, 108, 109, 571, 572, 583, 584 and/or 589).

## 3. Discussion

This study describes the molecular epidemiology of non-LGV *C. trachomatis* across a wide area in Spain during 2018 and 2019, using two genetic characterisation techniques, namely, *ompA* genotyping (n = 636) and MLST (n = 142), which both gave high yields (91% and 90%, respectively). Although genotyping highlighted the high genetic diversity of *C. trachomatis*, its discriminatory capacity (DI 0.791) was lower than that of MLST (DI 0.981), the latter technique therefore being more appropriate for the analysis of outbreaks, clusters and transmission networks [5]. As well as the great genetic diversity observed, *C. trachomatis* infection was found to be widely and evenly distributed across Spain, although with marked differences as a function of sex, and above all, sexual behaviour, certain genotypes and STs being more common among MSM.

Genotypes D–G and J represented 89% of the strains identified, and were detected in all the areas studied. Overall, as reported in most countries, genotype E was the most common (35%) [12,13,14]; however, the distribution of genotypes differed as a function of sex and particularly sexual behaviour. Indeed, differences in clinical and epidemiological characteristics of the populations included in different studies may influence the relative distribution of genotypes, and hence, help explain some of the differences reported [15,16,17,18,19,20,21,22,23]. Nonetheless, research carried out in geographically distant countries has found that genotypes D, G and J are predominant in MSM [3,24,25,26,27,28,29,30], as in our study, suggesting that this finding is robust. There is no consensus on whether this pattern is due to differences between genotypes in tissue tropism [31], or epidemiological network structures that facilitate the spread of some genotypes as a function of sexual behaviour [25,26,27,28,29,30].

Although genotypes I, H and K were found in a relatively small percentage of cases (10% overall), they were present in most of the areas studied. Their absence in two areas may be related to smaller numbers of samples having been collected there and the structure of the population analysed (MSM vs. MSW and women). Previous studies have also observed these genotypes in only a small percentage of cases, but with broad geographical distributions [13,21,23]. Interestingly, our study indicates a greater percentage of genotype I in women than in men, a pattern which, to our knowledge, has not been reported previously. Genotypes associated with trachoma (A–C), especially genotype B, have been sporadically detected in genital infections in trachoma-free countries [9,17,22,24,25,32,33]. It has been suggested that these cases could be the result of recombination between trachoma and sexually transmitted strains [7,8,9,34,35]. The fact that these genotypes are uncommon in genital infections, as in our study, in which only three strains were detected (all genotype B), suggests a low rate of transmission compared to that of more commonly found genotypes, attributable to a lower fitness for dealing with the host immune system or a more recent evolutionary origin [4,16].

The variety of *ompA* genotypes observed in this study between 2018 and 2019, and their distribution in the general population, as well as by sex and sexual behaviour, is similar to that found in a previous study carried out in Spain in 2011–2012, although that study had a more limited geographical scope [13]. We did not observe relevant changes in the distribution of genotypes over time, this remaining stable, which suggests that all the genotypes, including the less common ones, are in continuous circulation and spread across the population.

The wide range of multilocus STs, the high intra-genotype diversity and the high percentage of STs not previously described (29.9%) highlight the strong discriminatory power of MLST and its usefulness as a method for the molecular characterisation of *C. trachomatis* [36]. On the other hand, 7% of STs were shared by more than one *ompA* genotype, which supports the idea that recombination is common in this gene, and this, together with its lower diversity, is a limitation of *ompA* genotyping [5]. The greater discriminatory power of MLST allows for a more detailed study of infection transmission patterns based on clinical and epidemiological variables, such as sex, sexual behaviour and tissue tropism. Nonetheless, this typing method is laborious and is therefore difficult to implement in contexts outside of research.

Certain non-geographically clustered STs were more common. Previously, it has been shown that some STs are widely distributed across different geographical areas [36]. A better fitness or the development of specific transmission networks could explain the relative “success” of these STs and the differences between studies as a function of the population studied.

In our study, the most commonly detected STs differed between heterosexuals (ST30, ST35, ST97, ST100, ST128, ST148, ST276 and ST327) and MSM (ST33, ST52, ST108, ST109 and 571). Separate clusters for MSM and heterosexual populations have been previously observed in Sweden, the Netherlands, and the USA [28]. Our findings partially agree with those of a large study by Herrmann et al., analysing more than 2000 samples from 16 countries, in which the STs most commonly found in MSM were ST52, ST108 and ST109 (as well as ST58 in cases of LGV) [36]. On the other hand, Bom et al. found a lower genetic diversity in MSM than in heterosexual adults in Amsterdam [3], and we also obtained a lower DI in MSM than in MSW (0.947 vs. 0.977), suggesting greater clonality, although the between-group difference did not reach statistical significance, likely due to the relatively small group sizes. These results support the view that transmission patterns differ between these groups with separate sexual networks.

Regarding tissue tropism, we observed a different distribution of STs in rectal samples from women and MSM, consistent with the results of Versteeg et al., who, in a sample of 207 MSM and 185 women with anorectal infections, reported three distinct large clusters in each group [37], consistent with separate sexual networks. In line with this, analysing 120 rectal and 203 urine/urethral samples from MSM, Klint et al. did not find differences in the distribution of genotypes between these anatomical locations [27].

MLST also enabled us to identify potential genetic recombination between genotypes, having detected some identical STs in two different genotypes. Further, the identification of ST58 (most common in cases of LGV) in two samples containing genotype D suggests recombination between LGV and genotype D strains, or coinfection with strains of both genotypes. Cases of recombination of LGV with other genotypes have been reported previously in Spain [38,39].

This study has some limitations, in particular, the lack of clinical and epidemiological data for some cases. Secondly, although we analysed the distribution of genotypes in several different areas in the country that are relatively far apart, the results may not represent the situation in Spain overall. Moreover, the structure of the population in the areas included in the study was not completely homogeneous in epidemiological parameters such as sex and sexual behaviour, among others (Table 1), likely due to minor between-area differences in the structure of health services, but we believe that this would not have had a major impact on the validity of the overall results. Further, the *ompA* genotyping technique does not distinguish mixed infections with strains from other less abundant genotypes present in a sample, but mixed infections are not common. Finally, the selection of samples for the analysis of STs was not random, and hence, the relative weights of the most common STs should be interpreted with caution. Nonetheless, this study has allowed us to obtain a general picture of the circulation of STs in our country, their intra-genotype diversity and the existence of differences as a function of sexual practices.

In conclusion, in this study on the genetic characterisation of *C. trachomatis* in Spain, we observed good reliability of MLST, with this typing method having greater discriminatory power than *ompA* genotyping. The genotypes and STs identified were genetically diverse, and their distribution was geographically even and stable over time, though there were differences related to sex, and in particular, sexual behaviour. Despite the high diversity of STs, only a few were commonly observed and widespread across Spain, suggesting greater fitness. Regarding differences as a function of sexual behaviour, the strains most commonly found in MSM were from the genotypes/STs D/109, D/571, G/52, G/33 and J/108, these being different and less diverse than those detected in MSW. The differences between MSM and women in STs detected in rectal samples suggest that epidemiological factors have a stronger influence on transmission dynamics than tissue tropism, though this should be studied further with larger samples. All our findings indicate that *C. trachomatis* infection has distinct transmission networks in MSM and heterosexual populations in Spain.

## 4. Materials and Methods

### 4.1. Study Population and Design

We conducted a multicentre, observational, descriptive, cross-sectional study of the molecular epidemiology of *C. trachomatis* in 2018–2019 in seven tertiary hospitals in six areas in Spain: Andalucia (Valme Univeristy Hospital, Seville), Aragón (Miguel Servet University Hospital, Zaragoza), Asturias (Central University Hospital of Asturias, Oviedo and Cabueñes University Hospital, Gijón), the Balearic Islands (Son Espases University Hospital, Palma de Mallorca), Catalonia (Vall d’Hebrón University Hospital, Barcelona) and the Basque Country (Donostia University Hospital, San Sebastián), with a total catchment population of 3.050 million people, including 2.078 million aged between 15 and 65 years (Appendix A). The participating hospitals are the centres responsible for the microbiological diagnosis of STIs in their corresponding health regions, with most patients being identified through STI clinics, emergency departments, gynaecology units or family physicians. As part of the diagnostic process, samples are taken from patients who seek medical advice for suspected STI, both those who are symptomatic and those who are asymptomatic, but have engaged in high-risk sexual practices (unprotected sex) with a microbiologically confirmed case. Samples were also obtained in screenings carried out during pregnancy or HIV pre-exposure prophylaxis, but this origin was infrequent, since the implementation of screenings to detect *C. trachomatis* is limited in Spain. Each hospital collected leftover material from the first clinical sample of the week in which non-LGV *C. trachomatis* had been detected. If the detection method used was real-time PCR, only samples with an amplification cycle <35 were included in the study. All the samples were stored at −80 °C (anatomical site of sampling can be seen in Appendix A).

Subsequently, samples were sent to the Microbiology Department at Donostia University Hospital, where nucleic acids were extracted for *ompA* genotyping, and after this, multilocus sequence typing was performed in a subset of samples selected arbitrarily to cover all the different *ompA* genotypes detected, including cases from both sexes, with different sexual practices and with infection at different anatomical sites. 

### 4.2. Diagnostic Techniques

Each hospital used a commercial nucleic acid amplification technique for detecting *C. trachomatis* (Anyplex™ II STI-7 Detection, Seegene, Seoul, Republic of Korea; cobas^®^ CT/NG on a cobas^®^ 6800 system, Roche Molecular Systems, South Branchburg, NJ, USA; Versant^®^ CT/GC DNA 1.0 Assay, Siemens Healthcare Diagnostics, Erlangen, Germany; RealTime CT/NG Assay, Abbott, Desplaines, IL, USA; and Aptima^®^
*Chlamydia trachomatis* Assay, Hologic, IL, USA). These techniques allow for amplification using specific primers and probes for *C. trachomatis*, and in the case of PCR techniques, also other STI-related microorganisms (multiplex methods). Infection by LGV was detected using a specific PCR that amplifies a fragment of the polymorphic membrane protein H (*pmpH*) gene [40], and these cases were excluded from the subsequent analysis.

Nucleic acids were extracted using the EMAG^®^ automated nucleic acid extraction system (bioMérieux, Marcy l’Etoile, France). For *ompA* genotyping, we used a conventional PCR system to amplify the 990 bp fragment of interest [15], with subsequent Sanger sequencing of the amplicons obtained (3130XL Genetic Analyzer, Applied-Biosystems, CA, USA), and analysis of the sequences with the Basic Local Alignment Search Tool (http://www.ncbi.nlm.nih.gov/blast/Blast.cgi, accessed on 4 April 2023). For sequence typing, we used MLST, amplifying five highly variable genes (*hctB*, CT058, CT144, CT172 and *pbpB*), followed by bidirectional sequencing [5]. Subsequently, STs were assigned with tools available through the www.pubmlst.org website that uses the Uppsala University database.

### 4.3. Statistical Analysis

The distributions of the genotypes, as a function of the variables studied, were compared using the chi-square and Fisher’s exact tests. Values of *p* ≤ 0.05 were considered significant. Statistical analysis was performed using the SPSS statistics software (version 23, IBM, Chicago, IL, USA). To measure the genetic diversity based on the genotyping and sequence-typing data, we used the Simpson’s Diversity Index (DI). This index is considered an indicator of the diversity of qualitative variables, with values > 0.95 being considered highly desirable for molecular typing methods [41].

### 4.4. Ethical Considerations

The study was approved by the Clinical Research Ethics Committee of Donostia University Hospital (minutes 10/2017). Data on clinical/epidemiological characteristics (including symptoms, age, sex, sexual behaviour, and nationality) and microbiological information (including sample type, number of amplification cycles, genotypes, and STs) were recorded in a database, which was anonymised for subsequent analysis.

## Figures and Tables

**Table 1 ijms-24-06941-t001:** Clinical and epidemiological characteristics of the 636 cases of infection by *Chlamydia trachomatis*, in which the *ompA* genotype was obtained (2018–2019).

Population	Age (Years)		Sex/Sexual Behaviour			Nationality		STI Symptoms		
n (%)	<25	≥25	Woman	MSW	MSM	Spanish	Foreign-Born	Yes	No	Total
Asturias	86 (38.7)	**136 (61.3)**	136 (61.3)	63 (28.4)	23 (10.4)	179 (80.6)	43 (19.4)	**178 (80.2)**	40 (19.8)	222
Barcelona *	33 (33.7)	65 (66.3)	38 (38.8)	24 (24.5)	**34 (34.7)**	16 (55.2)	**13 (44.8**)	57 (59.4)	**39 (40.6)**	98
Gipuzkoa	37 (34.9)	69 (65.1)	40 (37.7)	**58 (54.7)**	8 (7.5)	86 (81.1)	20 (18.9)	77 (72.6)	29 (27.4)	106
Mallorca *	30 (36.6)	52 (63.4)	34 (41.5)	31 (37.8)	12 (14.6)	55 (67.9)	**26 (32.1)**	66 (81.5)	15 (18.5)	82
Seville	11 (28.9)	27 (71.1)	10 (26.3)	9 (23.7)	**19 (50.0)**	**37 (97.4)**	1 (2.6)	11 (28.9)	**27 (71.1)**	38
Zaragoza	36 (40.0)	54 (60.0)	43 (47.8)	30 (33.3)	17 (18.9)	64 (71.1)	26 (28.9)	**77 (85.6)**	13 (14.4)	90
Total	233 (36.6)	403 (63.4)	301 (47.3)	215 (33.8)	113 (17.8)	437 (77.2)	129 (22.8)	466 (73.6)	167 (26.4)	636

STI: sexually transmitted infection; MSW: men who have sex with women; MSM: men who have sex with men. In bold, *p* < 0.05 (chi-square or Fisher’s exact test as appropriate). * Data were missing on nationality, sexual behaviour and/or symptoms for some cases.

**Table 2 ijms-24-06941-t002:** Distribution of *ompA* genotypes in 636 cases of *C. trachomatis* infection in Spain (2018–2019).

Genotype	E	D	F	G	J	I	H	K	B	Total
n (%)										
Area:										
Asturias	85 (38.3)	41 (18.5)	39 (17.6)	22 (9.9)	12 (5.4)	12 (5.4)	4 (1.8)	4 (1.8)	3 (1.4)	222
Barcelona	26 (26.5)	19 (19.4)	15 (15.3)	16 (16.3)	9 (9.2)	10 (10.2)	1 (1.0)	2 (2.0)	0	98
Gipuzkoa	41 (38.7)	18 (17.0)	16 (15.1)	8 (7.5)	10 (9.4)	6 (5.7)	2 (1.9)	5 (4.7)	0	106
Mallorca	27 (32.9)	14 (17.1)	16 (19.5)	8 (9.8)	7 (8.5)	8 (9.8)	2 (2.4)	0	0	82
Seville	9 (23.7)	**12 (31.6)**	6 (15.8)	6 (15.8)	2 (5.3)	0	3 (7.9)	0	0	38
Zaragoza	37 (41.1)	15 (16.7)	14 (15.6)	10 (11.1)	7 (7.8)	4 (4.4)	1 (1.1)	2 (2.2)	0	90
Age (years):										
<25	86 (36.9)	35 (15.0)	46 (19.7)	22 (9.4)	13 (5.6)	19 (8.2)	6 (2.6)	5 (2.1)	1 (0.4)	233
≥25	139 (34.5)	84 (20.8)	60 (14.9)	48 (11.9)	34 (8.4)	21 (5.2)	7 (1.7)	8 (2.0)	2 (0.5)	403
Sex:										
Woman	114 (37.9)	42 (14.0)	**65 (21.6)**	24 (8.0)	19 (6.3)	**26 (8.6)**	6 (2.0)	5 (1.7)	0	301
Man	111 (33.1)	**77 (23.0)**	41 (12.2)	**46 (13.7)**	28 (8.4)	14 (4.2)	7 (2.1)	8 (2.4)	3 (0.9)	335
Sexual behaviour:										
MSW	**93 (43.3)**	35 (16.3)	**36 (16.7)**	13 (6.0)	11 (5.1)	12 (5.6)	6 (2.8)	7 (3.3)	2 (0.9)	215
MSM	16 (14.2)	**39 (34.5)**	4 (3.5)	**33 (29.2)**	**17 (15.0)**	1 (0.9)	1 (0.9)	1 (0.9)	1 (0.9)	113
*na*	*2 (28.6)*	*3 (42.9)*	*1 (14.3)*	*0*	*0*	*1 (14.3)*	*0*	*0*	0	*7*
Nationality:										
Spanish	162 (37.1)	85 (19.5)	68 (15.6)	45 (10.3)	31 (7.1)	26 (5.9)	7 (1.6)	11 (2.5)	2 (0.5)	437
Foreign born	39 (30.5)	24 (18.8)	27 (21.1)	14 (10.9)	7 (5.5)	10 (7.8)	5 (3.9)	1 (0.8)	1 (0.8)	128
*na*	*24 (33.8)*	*10 (14.1)*	*11 (15.5)*	*11 (15.5)*	*9 (12.7)*	*4 (5.6)*	*1 (1.4)*	*1 (1.4)*	*0*	*71*
STI symptoms:										
Yes	174 (37.3)	80 (17.2)	81 (17.4)	45 (9.7)	29 (6.2)	33 (7.1)	10 (2.1)	11 (2.4)	3 (0.6)	466
No	51 (30.5)	38 (22.8)	25 (15.0)	25 (15.0)	17 (10.2)	6 (3.6)	3 (1.8)	2 (1.2)	0	167
*na*	*0*	*1 (33.3)*	*0*	*0*	*1 (33.3)*	*1 (33.3)*	*0*	*0*	*0*	*3*
Total	225 (35.4)	119 (18.7)	106 (16.7)	70 (11.0)	47 (7.4)	40 (6.3)	13 (2.0)	13 (2.0)	3 (0.5)	636
95CI%	31.8–39.2	15.9–21.9	14.0–19.8	8.8–13.7	5.6–9.7	4.7–8.5	1.2–3.5	1.2–3.5	0.2–1.4	

STI: sexually transmitted infection; MSW: men who have sex with women; MSM: men who have sex with men; na: not available, in italics. In bold, *p* < 0.05 (chi-square or Fisher’s exact test as appropriate).

**Table 3 ijms-24-06941-t003:** Distribution of sequence types (STs) found in 142 selected samples from cases of *Chlamydia trachomatis* infection in Spain (2018–2019).

Genotype	ST	n/ST
E	**59, 327**	3 × 2
	3, 497	2 × 2
	56, 64, 147, 154, 573, 586	1 × 6
D	**35**	6 × 1
	**109**	4 × 1
	**571**	4 × 1
	58	2 × 1
	12, 90, 194, 243, 420, 585, 587	1 × 7
F	**148**	5 × 1
	**90**	3 × 1
	12	2 × 1
	85, 110, 231, 572	1 × 4
G	**52**	10 × 1
	**33**	5 × 1
	128	3 × 1
	28, 589	2 × 2
	27, 56, 265, 275, 279, 301, 575, 578, 582	1 × 9
J	**108**	5 × 1
	104, 136, 576	2 × 3
	112, 157, 267, 583, 584, 588	1 × 6
I	**276**	7 × 1
	**100**	6 × 1
	274	2 × 1
	130, 275, 517, 581	1 × 4
H	**97**	6 × 1
	443, 574, 580, 590	1 × 4
K	**30**	4 × 1
	380	2 × 1
	34, 358, 577, 579	1 × 4
B	108, 208	1 × 2
**Total**		142

In bold, the most common STs; in blue, STs found in more than one genotype. Sequence type numbers 571–590 correspond to new STs described in this study.

## Data Availability

Data on clinical/epidemiological characteristics and microbiological information were recorded in a database, which was anonymised for subsequent analysis.

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
