# Peer review of "Genetic Characterization of Non-*Lymphogranuloma venereum Chlamydia trachomatis* Indicates Distinct Infection Transmission Networks in Spain"

_ijms, 2023, doi:10.3390/ijms24086941_

Round 1

Reviewer 1 Report

I do not have comments it is very well written manuscript.

Author Response

Answer to Reviewer 1

Thank you very much for your kind comments.

Reviewer 2 Report

The aim of the manuscript was to assess the dynamics of the transmission of this infection, analysing the distribution of circulating ompA genotypes and multilocus sequence types of C. trachomatis in Spain as a function of clinical and epidemiological variables.

This was done by analyzing C. trachomatis genotypes of almost 700 C. trachomatis positive specimens with ompA PCR and Sanger sequencing and sequence types of  158 positive specimens with MLST.

Samples were collected from six areas. It is mentioned that patients were either “symptomatic or asymptomatic and engaged in high-risk sexual practices with a microbiologically-confirmed case.” (lines 81-83).  Knowing that a majority of infections can be asymptomatic, I wonder why asymptomatic individuals without such history were excluded? How were high-risk sexual practices defined?

What sample types were included (FVU, Vag, Cx, R, Ocular)? Rectal samples are mentioned on line 192.

There were 6 study sites, the samples were collected during 2 years and “material from the first clinical sample of the week in which non-LGV C. trachomatis had been detected” Is this description accurate, as they still had 698 specimens?

The description in lines 87-91 is quite unclear. “in all cases” and “a subset of samples”?

Diagnostic techniques (2.2.).: Several commercial test systems are listed. 

Were they all used at all sites?

They do not all contain internal controls.

They do not all give out data by amplification cycles or the Ct values. How were the samples then selected?

Table 1. It is confusing when the % of women is based on the total number of individuals, but the % of MSW and MSM is based on the total number of males.

Lines 139-140: “We detected the eight genotypes known to cause non-LGV infection (genotypes D-K)”. This is quite evident (see the description of C. trachomatis types). In fact, there were 9 types  (also B, but rarely). Genotype K was not detected at two locations, but they were the ones with the smallest sample sizes.

The findings: ompA genotypes were pretty much what has been reported earlier from Spain and from other countries. This data should be presented in a more condensed form.

The MLST typing produced some new information, including the novel ST types. This tells me that C. trachomatis strains are much more heterogeneous than have been previously though and that they are not genetically so stable as their intracellular lifestyle might predict.

The difference between MSW and MSM vs. females has been pointed earlier by Kohone et al Sex Transm Infect. 2012 Oct;88(6):465-9. As MSW and MSM obviously belong to different transmission chains, this kind of finding is to be expected.

The discussion should be condensed.

Reviewer 3 Report

Comments and suggestions

Title: Genetic characterization of (non-Lymphogranuloma venereum) Chlamydia trachomatis indicates distinct infection transmission networks in Spain

ID: ijms-2258635

General comment: This work has 2 objectives: 1) To analyze the ompA genotypes of non-LGV C. trachomatis to determine their geographical distribution in Spain and based on certain epidemiological characteristics of the population (including age, sex, and sexual behavior). , as well as genotypic changes over time; and 2) Perform MLST on a selection of samples that encompasses all genotypes, considering the aforementioned variables, and analyze the potential relationship of sequence types (STs) with transmission dynamics and tissue tropism. The study is interesting from the point of view of molecular epidemiology and at the local level the characterization of the pathogen is important for public health and to establish more effective control measures based on the knowledge of circulating pathogens.

Title: The title should not have parentheses

Summary Section: indicate the objective of the research in this section.

Indicate the unit of analysis (study population)

Introduction section: Describe perhaps in a couple of sentences what the MLST consists of and how useful it is in the focus of this research.

Methodology section:

The sampling procedure or the body regions from which the samples were obtained have not been described.

Explain why a control group has not been considered in this study. Especially if they have considered asymptomatic patients. This is a major limitation of the work.

Since among the objectives is to know the geographical distribution of the genotypes found, a map is strongly recommended.

Results section:

Explain why the age division of the population is 25 years.

It is not clear to me what was the selection criteria of the samples to carry out the MLST. If they used sexual behavior for this, wouldn't it be a selection bias?

Conclusions section:

In general it is well written, but it should stick more to the objectives set.

Round 2

Reviewer 3 Report

no